

# A machine learning based framework for IoT devices identification using web traffic

Sajjad Hussain[1], Waqar Aslam[2], Arif Mehmood[2], Gyu Sang Choi[3] and Imran Ashraf[3]

[1] Department of Information Security, The Islamia University of Bahawalpur, Bahawalpur, Pakistan
[2] Department of Computer Science & Information Technology, The Islamia University of Bahawalpur, Bahawalpur, Pakistan
[3] Information and Communication Engineering, Yeungnam University, Gyeongsan, Republic of Korea

## ABSTRACT

Identification of the Internet of Things (IoT) devices has become an essential part of network management to secure the privacy of smart homes and offices. With its wide adoption in the current era, IoT has facilitated the modern age in many ways. However, such proliferation also has associated privacy and data security risks. In the case of smart homes and smart offices, unknown IoT devices increase vulnerabilities and chances of data theft. It is essential to identify the connected devices for secure communication. It is very difficult to maintain the list of rules when the number of connected devices increases and human involvement is necessary to check whether any intruder device has approached the network. Therefore, it is required to automate device identification using machine learning methods. In this article, we propose an accuracy boosting model (ABM) using machine learning models of random forest and extreme gradient boosting. Featuring engineering techniques are employed along with cross-validation to accurately identify IoT devices such as lights, smoke detectors, thermostat, motion sensors, baby monitors, socket, TV, security cameras, and watches. The proposed ensemble model utilizes random forest (RF) and extreme gradient boosting (XGB) as base learners with adaptive boosting. The proposed ensemble model is tested with extensive experiments involving the IoT Device Identification dataset from a public repository. Experimental results indicate a higher accuracy of 91%, precision of 93%, recall of 93%, and F1 score of 93%.

## INTRODUCTION

The Internet of Things (IoT) has gained fame on the global level, and the use of IoT devices has become an essential part of our daily life (*Minoli, Sohraby & Occhiogrosso, 2017*). With the inception of smart homes, offices, and cities, increasing use of IoT devices is observed which is further expected to grow in the near future. The IoT paradigm provides ease of access, remote control, and 24/7 access to devices that have been used in a large number of applications. However, such massive use of IoT devices is not without its demerits, and

Corresponding authors
Gyu Sang Choi, castchoi@ynu.ac.kr
Imran Ashraf, imranashraf@ynu.ac.kr

data privacy and confidentiality concerns are high. As a result, IoT device identification has become an important element of data and network security. Different approaches have been designed in this regard. To identify the internet-connected devices, an effective machine learning algorithm is proposed during the periodic flow of features that are extracted during the communication (*Zhang, Gong & Qian, 2020*). Two ways of capturing the flow of packets on the network include active and passive capturing. The passive method is becoming more popular as compared to the active method as it does not use any additional software (*Bao, Hamdaoui & Wong, 2020*; *Maurice et al., 2013*).

The IoT provides such use of the network that would not have been possible otherwise. Using the physical layer, IoT devices can be identified by using novel deep learning frameworks (*Liu et al., 2020*). With the passage of time, various types of security issues are faced as the use of IoT devices becomes explosive. Network administrators use detailed information on devices on the network of smart homes or offices and control them accordingly. Many studies are based on detecting the unknown devices and known devices connected to the internet, improving the performance of such methods to identify the IoT devices is always needed (*Imamura et al., 2020*).

The growth of IoT devices invites many vendors that introduce new IoT devices without specific content. This increased the vulnerabilities for IoT devices and gave opportunities to intruders for active attacks on the IoT device. A wide range of IoT devices is designed for use in smart homes. Management becomes easy when data is analyzed, but it is becoming difficult in the case of a heterogeneous environment. IoT devices produce a lot of data that increases the chance of errors and missing data. A well-engineered system becomes smarter with computing capabilities and IoT devices with a lack of computing capabilities may be segregated as unprotected devices and restricted traffic may be provided to a device regarding its communication needs. A novel approach for IoT device identification based on the locality-sensitive mixture of traffic flow between user and target devices needs no feature extraction as required in the machine learning approach.

Cyberspace has increased since the introduction of newly developed IoT devices. It is very difficult to find a device that is not connected to the internet. Wearables, such as smartwatches, fitness trackers, glasses, medical devices, and other house appliances connected with any network are increasing day by day. It is also assumed that in this decade these devices will lead our lives as every 80 s one new IoT device is connecting to the internet (*Kelly, 2015*). Wearable devices are expected to be used for different tasks; such devices will be authorized for legal credential accessing systems and networks. However, it will open new types of insider and outsider intruder threats for leaking valuable information using cyber attacks (*Aksu, Uluagac & Bentley, 2018*). A network traffic flow model and its vulnerable structure can uncover the user's personal information to hackers. Good and well-planned communication increases the importance of IoT-based systems (*Ullah & Mahmoud, 2021*).

For building a smart home environment, devices used in the smart home should be identified using the same identification system by the different IoT platforms *e.g.*, machine to machine (M2M), global standard 1(GS1), etc. By focusing on the interoperability of heterogeneous IoT platforms, this issue can be solved by a conceptual identification

translator called a device name system (DNS). IoT devices used in the smart home are built by different global companies like Google, IBM, Intel, etc., as well as, by domestic companies like Samsung in Korea, LG, *etc*. IoT devices used in the smart home should be registered and monitored by some mobile application so that they can be controlled *via* the internet. By adopting some translator between heterogeneous platforms, the performance of the identification system can be increased (*Koo & Kim, 2017*).

An unknown IoT device connected to the smart home network may be vulnerable and could be used for internal attacks. Hence, identified IoT devices secure the network and are easy to monitor (*Li et al., 2013*). IoT devices with limited computational capabilities and with high traffic can restrict the identification process. The genetic algorithm(GA) can be used to identify IoT devices, as GA uses unique features and reduces selected features by the use of different approaches to obtain the best results (*Aksoy & Gunes, 2019b*). IoT device type identification (DTI) has also become more popular for monitoring and managing the entire network containing a large number of IoT devices and wearables. Deep learning is also used for the classification or identification of different types of IoT devices. Devices type identification is associated with large computational complexities and model sizes (*Qing et al., 2020*). For unknown device identification, the response time should also be considered (*Zhu et al., 2021*).

A variety of devices connected to the internet such as printers, phones, and network-connected cameras are commonly used with the help of the internet. As the increase in the use of IoT devices is increasing, strict rules need to be defined to ensure their security. Identification of devices is useful for specifying the traffic patterns in normal communication (*Kawai et al., 2017*). Different vendors are producing IoT devices without specific configurations and design specifications which may lead to increased vulnerability. In most attacks related to IoT, *e.g.*, Mirai, attackers try to control and manage the devices by performing abnormal behaviors. Repaid growth in the use of IoT, and security-related concerns of IoT devices have come to light. A large number of vendors are producing IoT devices connected to the network by getting IP addresses for homes and offices having no specific designs and security constraints. Therefore, to identify IoT devices, and detect malicious behaviors and attacks device usage description model is helpful to mitigate such risks (*Wang et al., 2020*). There are security laps in such networks with vulnerable devices having no security mechanisms. Securing networks in the presence of such vulnerable devices requires some power tool to identify the newly connected device for vulnerability. Such vulnerabilities should be dealt with properly and promptly. However, often many vendors are unable or do not want to provide patches to remove such vulnerabilities. Most IoT users do not use these devices properly with outdated software; some of them even do not know the proper use and updation of security mechanisms which may further aggravate the security problems.

For IoT devices, identification is necessary for end users to manage the smart home work to reduce manual tasks in case of a higher number of IoT devices are connected. Supervised machine learning models can distinguish between different IoT devices based on network flow features, and training data with known classes or labels can be used to train and identify new connected devices (*Ammar, Noirie & Tixeuil, 2020*). The machine

learning model is always re-trained due to various reasons, whenever a new device is added, environmental changes, or due to any update in existing security software. The performance of the machine learning model depends upon some of the given factors, such as feature engineering, model selection, parameter tuning, and representation of training data (*Charyyev & Gunes, 2020*). Feature selection is also important, but it requires expertise to select features among the number of given features. It is also important as feature selections reduce the computational cost, and storage consumption and also prevent the model from overfitting and underfitting on the training data (*Aksoy & Gunes, 2019b*).

This study presents an accuracy-boosting model for IoT device identification that utilizes a random forest (RF) and extreme gradient boosting (XGB) model. In addition, several well-known machine learning models are employed to have a performance appraisal. Logistic regression (LR), support vector classifier (SVC), AdaBoost (AdB) classifier, k-nearest neighbor (KNN), decision trees (DT), gradient boosting machine (GBM), XGB, and RF are implemented in the current study. In addition, experiments are performed regarding the use of different feature extraction approaches and analyzing their impact on classification accuracy. Constant features, Chi-square (Chi2) features, information gain-based features, and highly correlated features are utilized for experiments. Random stratified k-fold cross-validation is used to validate the performance of the proposed approach.

This article is organized as follows: we review related works in 'Related Work'. Next, we introduce the proposed methodology in 'Proposed methodology'. In 'Results and discussion', we present experimental results and discussions for IoT device identification through web traffic. Finally, we conclude this work in 'Conclusion'.

## RELATED WORK

There are many studies related to IoT device identification. Based on information gathered from the network, the similarity of behavior helps to identify the devices on the network. By using a special technique called the weight of identification methods multiple devices in the network based on similarity can be identified more accurately rather than other methods in this field. By combining the results of multiple identification methods, accuracy can be improved as compared to individual methods (*Imamura et al., 2020*).

*Kawai et al. (2017)* proposed a method to identify IoT devices using the packet distribution and communication time interval. The authors identified nine kinds of devices, some of them having the same type but from different vendors. An average accuracy of 88.3% is obtained using a support vector machine (SVM). Some of the devices showed an accuracy of 69.8% which is worse than overall identification accuracy. It is very difficult to get enough traffic for identification as such kinds of devices generate less traffic.

### Fingerprinting

Network fingerprinting (*Vanaubel et al., 2013*; *Kohno, Broido & Claffy, 2005*) is a technique that is used to detect network-connected devices by observing the flow of traffic received and sent by IoT devices in the entire network. Initially, fingerprinting was implemented along with operating systems where fingerprinting was used for analyzing the packets to

check which specific operating system (OS) is operative on the device and to conclude which application and firmware is running on the system. For the occurrence of any activity, the network analyzer presented in *Lyon (2016)* is an open-source fingerprinting tool. Network mapper uses IP traffic to check which OS version is running on the host and how many hosts are present on the network. Active and passive fingerprinting are the two main types of fingerprinting that are used so far. In some cases, active and passive fingerprinting are combined to obtain better results.

## Active device fingerprinting

An IoT device can be identified by its response time during communication. *Bratus et al. (2008)* proposed an active fingerprinting approach for discovering the firmware of wireless devices by analyzing its response. *Sieka (2006)* proposed an active fingerprinting method that discovers wireless end devices by analyzing the response time of how devices communicate and respond. Active fingerprinting provides additional details about connected IoT devices. It depends on the accessibility of devices and also informs about additional network traffic. However, it can be a bottleneck for some measurable platforms. Using active fingerprinting, new devices have to be identified directly before establishing a connection with the end device (targeted device).

## Passive device fingerprinting

Passive fingerprinting analyzes the communication of connected devices, extracts information from the traffic, and creates a unique baseline behavior for each device. This method does not require establishing a connection with the targeted device and does not create artificial traffic on the network.

*Meidan et al. (2017)* were the first to introduce a multi-stage classifier by analyzing networks to identify IoT devices. Identification requires collecting network traffic from local devices such as laptops, smartphones, etc. in the form of packet capture files. Transmission control protocol (TCP) packets are combined during the communication process and represented by a feature vector. This approach also considers specific features such as IP addresses and source or destination interacting ports. However, these features are not highly related to showing the behavior of the specific device and can not select a generic approach to model the device activity.

## Hybrid device fingerprinting

This approach works in two steps: first (passive fingerprinting) collects information about network traffic and starts to examine device behavior. Then, the second step initiates communication between connected devices to assess the results of the first step and check the missing information, and deal with it. *Gagnon & Esfandiari (2012)* proposed a hybrid approach for OS discovery.

Device types and classification can be carried out using deep learning-based DTI but it requires more computational processing capabilities and large model sizes. The use of a lightweight convolutional neural network (CNN) reduces model sizes by removing redundant fully connected layers and replacing common convolution with separable

convolution. Thus, it can have less computational complexity and model size of CNN for device type identification (*Qing et al., 2020*).

Wearable gadgets, smart watches, medical devices, and fitness trackers may have authorized credentials or may cause a risk of information leakage. Different machine learning algorithms are used in the training process and out of these only the best-performing algorithms can be selected for the testing phase. By performing testing on real wearable devices connected with the network using Bluetooth protocol, machine learning fingerprinting provides reliable cyber attack intelligence with an average 98.5% precision and 98.3% recall for identification of wearable using Bluetooth classic protocol (*Aksu, Uluagac & Bentley, 2018*).

IoT SENTINEL is a system to identify the type of newly added devices and enforcement of rules to enable constraints on the communication of vulnerable devices so that damage to results can be minimized. Type identification is based on the communication between devices; IoT SENTINEL has minimal performance elevation. A few devices were tested with this technique with considerable results but some devices did not respond due to software version variations (*Miettinen et al., 2017*). Identification of IoT devices with method locality-sensitive IoT fingerprinting (LSIF) relies on the hash of the traffic flow, without depending on feature extraction from traffic flow and parameter tuning, *etc*. The performance of LSIF was much better related to other methods due to its lightweight architecture. It is suitable for a large network in the online identification of devices (*Charyyev & Gunes, 2020*).

A cross-layer protocol fingerprint can be used to identify the network-connected devices on a large scale instead of a single protocol fingerprint. Using a cross-layer protocol fingerprint, first, a scheme is designed to collect cross-layer packets of HTTP and TCP. The distinct aspects of HTTP and TCP protocols are selected to establish the discrepancy and harmony of features to other field values. By utilizing CNN and long short-term memory network (LSTM) features specific fields of HTTP and TCP can be extracted and fingerprints of these fields can be constructed to succeed in the identification of IoT devices with high accuracy and better time efficiency (*Yu et al., 2020*).

With the unfolding of cyberspace matter surveillance, researchers are preparing to improve the way of identification of IoT devices. There are two ways of identification of IoT devices, one is passive and the second one is proactive identification. Using the passive method different aspects of communication that are connected with the same network can be used to effectuate the device's identification. In the area of wireless communication devices, security issues of the network in case of interruption and securing the decentralized composition of a ZigBee Ad-Hoc network, the special actual features of radio frequency between connected devices (RF-DNA) can be utilized as a fingerprint to carry out device identification (*Patel, Temple & Baldwin, 2014*).

Identification can also be carried out by identifying the model and types of devices and analyzing the communication based on network packets. It can be done by considering the likeness of features extracted from the own network packets. For example, a camera transfers video data continuously, whereas, the size of data transfer in the case of a temperature sensor may differ depending on the model and type of device. An individual

device can be identified by considering the flow of traffic. The identification of devices by analyzing the communication of devices is performed without specific tools. It can be done by appointing a special role for cameras and workplace devices as these are valid goals for device identifiers. Furthermore, the identification of models by analyzing the communication of the same type of devices by disclosing the connection between the information stored in the packet header is used for identification. *Noguchi, Kataoka & Yamato (2019)* performs identification of 11 different types of cameras using this pattern and successfully identified nine of them.

An IoT device identification method based on deep learning model CNN and bi-directional long short-term memory (BiLSTM), called end-to-end IoT identification device is investigated that performs better as compared to traditional techniques from the perception of accuracy. The IoT ETEI method can identify the IoT device with great accuracy on public datasets related to IoT devices, even if they are using some security protocols (*Yin et al., 2021*). IoTSpot (*Deng et al., 2019*) includes all possible features in the TCP header of each TCP stream, with a total of 19 features. On the other hand, the principal component analysis (PCA) algorithm is also used to choose the features that contribute more toward identification. In the end, a radiofrequency fingerprinting (RF) model is used to pick up precise network traffic signatures for the identification of devices.

Radiofrequency fingerprinting (RFF) is a specific feature related to the hardware of IoT devices that cannot be intruded on easily. Radiofrequency uses feature selection or dimensionality reduction algorithms such as robust principal component analysis (RPCA) while SVM can be used for classification. Both, theoretical modeling and trial verification are carried out. Reliability and diversity of RFFs are analyzed and estimated, whereas classification results are displayed in a real IoT-based environment (*Tu et al., 2019*). IoTFinder, a system that efficiently finds the IoT device on a large scale, supports scattered passive DNS data gathering. IoTFinder supports developing a machine learning-based system that aims at accurately finding a large number of IoT devices based on DNS fingerprints. It can find the devices, no matter where they are located, for example behind a NAT or whether an IPv4 or IPv4 is assigned. IoTFinder is a multipurpose classifier and can determine its accuracy in the number of various configurations whether data is collected from DNS traffic or some other resources such as an IoT traffic dataset. IoTFinder can find IoT devices more accurately, even those that are hosted with non-IoT devices in the same network or have mixed traffic (*Perdisci et al., 2020*).

IoT network is a scattered infrastructure as a large number of IoT devices connect with the same network and build a radical concentrated network. In heterogeneous environments where the sender and receiver have heterogeneous devices, collecting device network information, testing, and checking the approaches remains challenging. A device discovery tool using a series of experiments can collect and give information on IoT devices connected with different networks using different protocols. In such cases, different application layers and associated open ports are used and the analyst needs to analyze different event handlers. *Khan et al. (2020)* performs experiments to identify the IoT devices associated with open ports and IoT and non-IoT devices. In addition, device identification is very important in investigating and analysis of traffic on the application layer.

**Dataset Statistics**

| | |
|---|---|
| Number of Variables | 298 |
| Number of Rows | 1800 |
| Missing Cells | 0 |
| Missing Cells (%) | 0.0% |
| Duplicate Rows | 23 |
| Duplicate Rows (%) | 1.3% |
| Total Size in Memory | 4.2 MB |
| Average Row Size in Memory | 2.4 KB |
| Variable Types | Numerical: 162 <br> Categorical: 136 |

**Figure 1  Data statistics.**

## PROPOSED METHODOLOGY

This study utilizes web traffic for IoT device identification. Choosing the most relevant features for the given task is very important for obtaining better results using machine learning models. For our research work, first, we have to identify the device type. In addition, the features are obtained from the communication packets between network-connected devices and the gateway.

### Dataset

The dataset used in this article is taken from Kaggle and is publicly available at *AMI (2021)*. It comprises packets extracted from the traffic flow of IoT devices. It contains traffic flow information of nine IoT devices that have 1800 subtitles and 297 features, as shown in Fig. 1. 23 duplicate rows in the dataset were removed. Code enabler is used for textual (data given in target class) to numeric data. Feature selection techniques are used for preprocessing purposes to select suitable features.

It has a traffic flow of nine IoT devices including lights, smoke_detector, thermostat, motion_sensor, baby_monitor, socket, TV, security_camera, and watch. The data is split in the ratio 0.70 to 0.30 for training and testing, respectively. Various feature extraction methods are investigated to analyze their suitability for machine learning models. Figure 2 shows the distribution of data samples regarding device categories.

### Feature selection

The dataset contains a total of 298 features for each of the nine devices. Using different feature engineering techniques, we need to find the best-performing features from the whole data. Table 1 shows the names and descriptions of a few features.

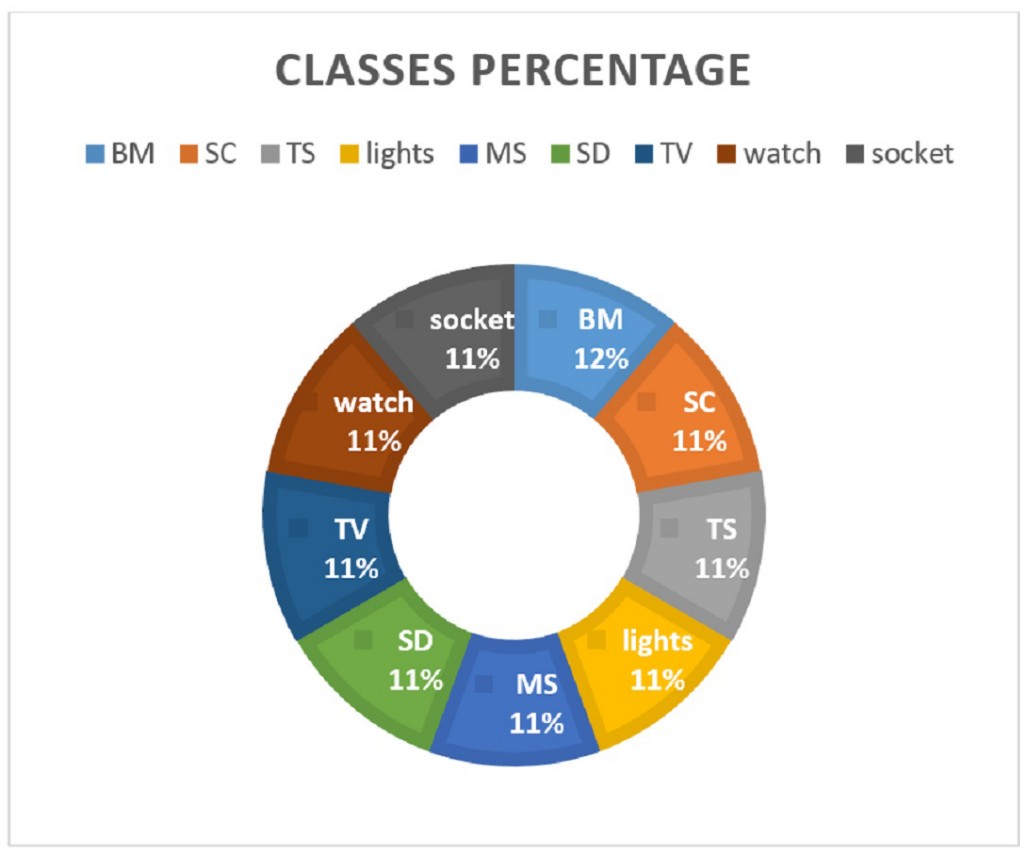

**Figure 2** Percentage of data records for each class.

### Constant features

Features that are actually not important in problem-solving are called constant features. They are also called zero variance features as they do not show any variance in data. This technique is applied only to independent elements of data ($X$) and not to the output ($Y$). By removing zero variance features (constant features), noisy 253 features were selected to obtain a higher accuracy for IoT device identification.

### Chi-squared test

This test computes the relationship between every non-negative feature and target values. It is used to estimate categorical variables in the classification task. The chi-squared test is used to choose n_features based on *p*-values and f-score. Having more f-score and lessor *p*-values of test chi-squared statistic from X and must contain only non-negative values. It also compares the classification of different classes of Y between different categories of features in the case of target classes. In the Chi2 method, the top 10 features having lesser *p*-values were selected for this study.

**Table 1  Features and their relevant description for the dataset.**

| Feature | Description |
|---|---|
| IP, port | IP and ports of client / server |
| Packets | Number of packets sent by client/server / both |
| Ack | Number of Ack packets sent by client/server / both |
| packetsAB_ratio | Ratio between packets sent by the client and sent by server |
| Asn_ | Number of autonomous systems served as a client, server |
| Push_ | Number of packets with PSH flag sent by client/server / both |
| Reset_ | Number of packets with RST flag sent by client/server / both |
| Bytes_ | Number of bytes sent by client/server / both |
| bytesAB_ratio | Ratio between the number of bytes sent and number of bytes received |
| Sslcountcertificates | Number of SSL certificates |
| Cap_date | Date of data capturing start |
| Sslcountclient | Client: Number of supported SSL cipher algorithms / ciphersuites / compressions /elliptic curves / key exchange algorithms / MAC algorithms |
| Sslcounserver_ | server: Number of supported SSL cipher algorithms / ciphersuites / compressions /elliptic curves / key exchange algorithms / MAC algorithms |
| Sslhandshakeduration_ | SSL handshake duration: Minimum value, quartile 1, average, median (quartile 2), sum, quartile 3, maximum value, standard deviation, variance, entropy |
| , …, | |
| Packetsize.. | Packets size: Minimum value, quartile 1, average, median (quartile 2), sum, quartile 3, maximum value, standard deviation, variance, entropy |
| Suffixis.. | Suffix of HTTP dominated host is one of recent top 4most frequent: com, net, etc. |

## Information gain

Mutual information computes the quantity of information one can attain from one random variable number. The mutual information between two random variables $X$ and $Y$ is given as follows

$$I(X:Y) = H(X) - H(X|Y) \tag{1}$$

where $I(X:Y)$ is the mutual information for $X$ and $Y$, $H(X)$ is the entropy for $X$ and $H(X|Y)$ is the conditional entropy for $X$ given $Y$.

Information gain helps to automatically select useful features to reduce the complexity but results may be affected negatively by selecting features using information.

## Correlation

Correlation refers to the dependence of different features on each other and features that are highly correlated with each other are selected for better results. If one feature is highly correlated with another one, we can use one of them and ignore the other ones. While

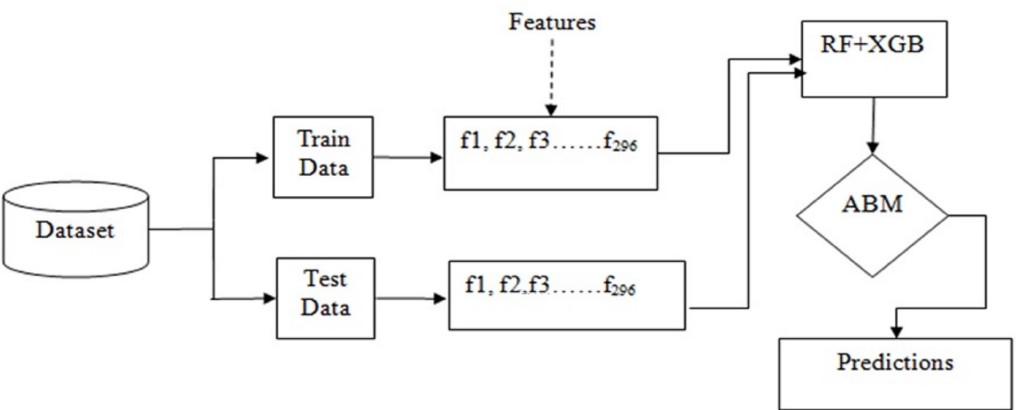

**Figure 3** Workflow of the adopted methodology.

using correlation, 129 selected features increased the performance of models. Results are degraded as we remove some features; if all features were used, results show higher accuracy.

## Models implementation

Basically, the problem of IoT device identification is a classification problem that can be solved using ensemble techniques such as bagging and boosting. Figure 3 shows the workflow of the adopted methodology for this purpose.

### Why ensemble model is chosen

Ensemble models is a machine learning technique to combine multiple models called weak learners or base estimators, to get better results. Single models traditionally face the following challenges

**High variance:** The model is very sensitive to given inputs to the learned features.

**Low accuracy:** For fitting a whole training data, one model is not enough to meet the expected results.

**Features noise and bias:** The model deliberately depends upon one or more features during the prediction of the results.

### Ensemble Algorithm

A single model can not predict more accurately for a given dataset. Machine learning models have restrictions and making a model with higher accuracy is challenging. If we construct a model by combining two or more models that can boost accuracy. This is used to aggregate the results of combined models to reduce model errors and keep generalizations. A term called meta-model is used.

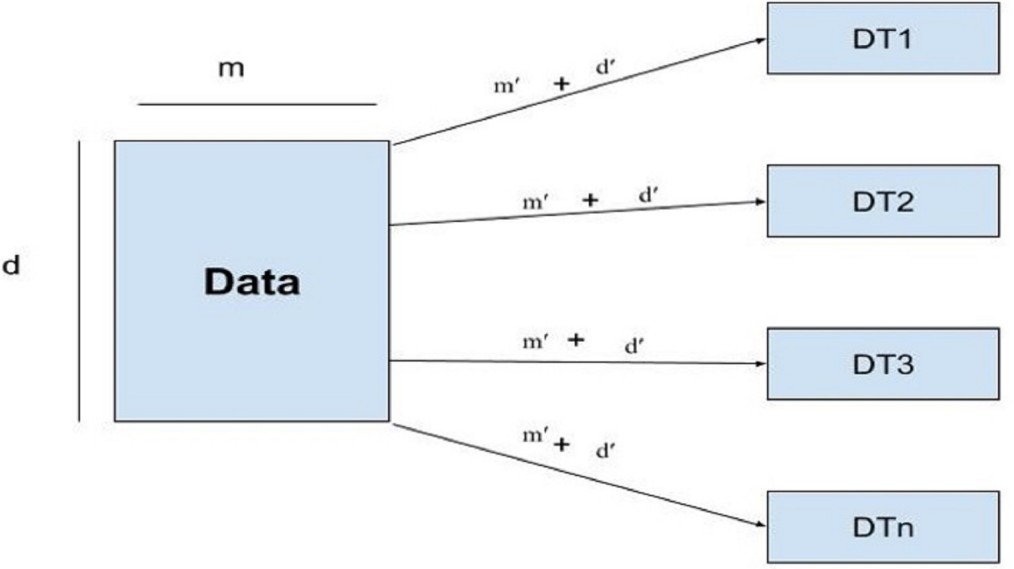

**Figure 4** Architecture of the RF model.

### *Ensemble techniques*
### Bagging

Bagging is the process of making the data available for training repeatedly. The dataset is divided into multiple subsets and each model learns the results of the error by the previously applied model using a slightly different subset of the training dataset. Bagging reduces the variance and minimizes overfitting. In the case of a classification problem, samples of the dataset are given to different models parallel for training. It provides accuracy scores for all models and their average can be taken to make the final accuracy score. On the other hand, for regression problems average output of the models is combined. Random forest is one of the best examples of this technique.

### Random forest

Random forest uses subsets of features called feature sampling (FS) and subsets of records called row sampling (RS) as subsets of training data to construct several divided trees, as shown in Fig. 4.

Each tree in the RF is independently trained as base learners. Whereas, in boosting, these learners are weak. Multiple decision trees are constructed to fit every training dataset. Row sampling with replacement is used randomly. For output/results, the majority voting classifier is used. As in this study classification of IoT devices is performed using a voting classifier. The given dataset is divided into row samples and feature samples; if features of the dataset are defined as, 'm' and m' represent features sampling whereas, rows are represented by 'd' row sampling as d' then RF will follow the given condition as

$$m' < m \& d' < d. \tag{2}$$

*Boosting*

The boosting technique works based on combining the models sequentially. As every model has some wrong predictions called weak learners, weak learners are combined to form strong learners.

## Gradient boosting

The gradient boosting model is a considerable method that has more predictive production. Xgboost and Catboost are approved boosting models that can be used for both regression and classification problems. Xgboost works on a similarity base, it counts similarity weight by creating binary decision trees.

## Adaptive boosting

This model is built on top of weak learners having simply restricted prediction abilities. Sequential trees are built on their weights of earlier knowledge of accuracy. It is performed in a sequential way instead of parallel. The model is trained on a subset of the dataset and tested. If errors occur, it is again trained and tested having knowledge of previous results. Error estimates are repeated for the given number of iterations. Overall weight should be equal to 1. If a tree comes to only one level then it will generate an error that the tree will act as a weak learner. Basically, Adaboost is used to classify image and text data more significantly and is less prone to overfitting.

## Accuracy boosting model (RF + XGB)

The accuracy boosting model for IoT device identification based on web traffic is a machine learning model that has been designed to identify nine types of IoT devices using their web traffic patterns. This model utilizes two powerful classifiers, the XGB and RF, to achieve a high level of accuracy and improve the performance of the model compared to other models. The XGB algorithm is utilized in this model to effectively handle high-dimensional and complex data. It uses gradient boosting to iteratively build a sequence of weak classifiers, each of which attempts to improve upon the errors of the previous one. This creates a strong ensemble of models that can accurately predict the type of IoT device based on web traffic patterns. The RF, on the other hand, is known for its ability to handle large datasets and high levels of noise. It creates multiple decision trees and combines them to obtain a more accurate and stable result. The randomization of samples and features ensures that the model is robust to different data distributions. By combining the strengths of both algorithms, the Accuracy Boosting Model for IoT device identification based on web traffic can achieve high levels of accuracy in a variety of experiments as discussed in the result and discussion session. It is specifically designed to identify nine types of IoT devices, such as smart watches, baby-monitor, and smart thermostats, based on their unique web traffic patterns. This model has been developed with default parameters that have been optimized for high accuracy. However, it can be further tuned to achieve even better results on specific datasets. Overall, the Accuracy Boosting Model is an excellent choice for anyone looking to accurately identify IoT devices based on their unique web traffic patterns. It has the potential to revolutionize the field of IoT device identification and improve the security of IoT devices.

## RESULTS AND DISCUSSION

In this part, we assess the performance of the proposed solution on the identification of IoT devices. First, we present the results of metrics to check the model's efficacy, and secondly, discuss the results of considered models and the proposed model.

A publicly available dataset, the IoT Device Identification dataset is utilized for experiments. It comprises packets extracted from the traffic flow of IoT devices. A dataset with 09 IoT devices that contained 1800 subtitles and 297 features was used. The 70% data was used for training the proposed model and 30% measurements of the data for testing purposes.

Confusion matrices are required to show the results between actual values and predicted values when handling classification problems. It gives overall results for recall, precision, and accuracy.

The accuracy of a model is the sum of true positive (TP) and true negative (TN) predictions divided by the sum of all elements of the confusion matrix:

$$Accuracy = \frac{Total\ True\ predictions}{Total\ correct\ and\ incorrect\ predictions} \tag{3}$$

The precision tells about how many are correctly predicted from all positive predictions.

$$Precision = \frac{TP}{TP + FP} \tag{4}$$

The recall shows how many results are correctly classified among true positive and false negative (FN) predictions.

$$Recall = \frac{TP}{TP + FN} \tag{5}$$

Where TP shows how many times IoT devices are correctly classified, TN shows how many times these are correctly identified related as not IoT devices. FP shows the number of occurrences incorrectly identified as IoT devices, and FN shows how many times IoT devices are classified as new IoT devices.

Two essential areas of machine learning are training and testing performed on the selected models. Before measuring accuracy scores we divide the data for the training set (70% of the data) and testing (30% of the data). In the first step, we used the training subset of the dataset to train machine learning models while the second step involved using the testing subset to evaluate the performance of trained models. Models we implemented using the scikit-learn Python library.

### Results using constant features

The first set of experiments is carried out using the constant feature selection method. For this purpose, constant features (zero variance features) are removed from the feature set. The models are trained and tested using selective features and results are provided in Table 2. The performance of the models is based on 253 features from the data which shows that the best accuracy is obtained by the RF. It is followed by the proposed approach ABM which obtains 90% accuracy for IoT device identification. It also obtains the highest CV score of 91%. XGB and GDB models also achieve 90% accuracy using constant features.

**Table 2  Performance of models using selective features by removing constant features.**

| Model | Precision | Recall | Accuracy | CV Scores |
|---|---|---|---|---|
| LR | 89 | 89 | 89 | 11 |
| SVC | 86 | 85 | 85 | 18 |
| AdB | 27 | 44 | 44 | 48 |
| KNN | 87 | 86 | 86 | 79 |
| DT | 88 | 88 | 88 | 87 |
| XGB | 90 | 90 | 90 | 90 |
| GDB | 90 | 90 | 90 | 90 |
| RF | 91 | 91 | 91 | 90 |
| ABM | 90 | 90 | 90 | 91 |

**Table 3  Performance of models using Chi2 features.**

| Model | Precision | Recall | Accuracy | CV Scores |
|---|---|---|---|---|
| LR | 15 | 21 | 21 | 11 |
| SVC | 15 | 21 | 21 | 18 |
| AdB | 28 | 26 | 26 | 25 |
| KNN | 46 | 44 | 44 | 48 |
| DT | 49 | 37 | 37 | 49 |
| XGB | 49 | 39 | 39 | 49 |
| GDB | 49 | 37 | 37 | 49 |
| RF | 49 | 37 | 37 | 49 |
| ABM | 57 | 47 | 47 | 50 |

## Results using Chi2 features

Chi2 is another important feature selection method to select important features from a set of all features. For this study, the features are selected based on lessor $p$-value. A total of 10 features are selected using lessor $p$-values. Table 3 shows the experimental results for all the employed models. However, the performance of the models is poor as compared to using constant features. The classification precision of all the models is drastically reduced. For example, the precision of LR and SVC is reduced from 89% and 86% to 15% each which shifted from constant features to Chi2 features. DT, XGB, GDB, and RF show comparatively better performance with 49% accuracy each. The proposed approach still managed to obtain a classification precision of 57% using Chi2 features. DT, XGB, GDB, and RF all have 49% CV scores while the proposed approach has a CV score of 50%.

## Results using information gain

Another feature selection method, called mutual information gain is also utilized to extract the top 10 most important features from the set of all features. Experimental results using information gain-based features are shown in Table 4 which indicates that the performance of the models is better than using Chi2 features but poorer than using constant features. Results of RF, GDB, and XGB show a CV score of 82% while the purposed model shows an 83% CV score which is higher than other models.

**Table 4  Performance of models using information gain-based features.**

| Model | Precision | Recall | Accuracy | CV Scores |
|-------|-----------|--------|----------|-----------|
| LR | 65 | 69 | 69 | 70 |
| SVC | 64 | 69 | 69 | 61 |
| AdB | 37 | 41 | 41 | 46 |
| KNN | 75 | 81 | 81 | 80 |
| DT | 73 | 79 | 79 | 80 |
| XGB | 75 | 81 | 81 | 82 |
| GDB | 75 | 80 | 80 | 82 |
| RF | 76 | 82 | 82 | 82 |
| ABM | 75 | 81 | 81 | 83 |

**Table 5  Performance of models using highly correlated features.**

| Model | Precision | Recall | Accuracy | CV Scores |
|-------|-----------|--------|----------|-----------|
| LR | 89 | 86 | 86 | 25 |
| SVC | 85 | 82 | 82 | 24 |
| AdB | 58 | 57 | 57 | 50 |
| KNN | 88 | 87 | 87 | 78 |
| DT | 87 | 86 | 86 | 87 |
| XGB | 93 | 93 | 93 | 90 |
| GDB | 91 | 91 | 91 | 90 |
| RF | 92 | 91 | 91 | 90 |
| ABM | 93 | 92 | 92 | 91 |

## Experimental results using high correlation-based features

In the given dataset the features have a different level of correlations. So, experiments are also carried out using the selective features based on high correlation and results are presented in Table 5. Results indicate that using high correlation-based features with the models, the results are better than using Chi2 and information gain. RF, GDB, and XGB show CV scores of 90% each while the proposed approach ABM shows a 91% CV score.

Since the proposed ABM model showed better results than other models used in this study, a performance comparison using different features is presented in Fig. 5. The comparison is shown only for ABM with all the features used in this study.

## Validation using additional dataset

To further validate the performance of the proposed model, we carried out additional experiments using the IoT Dataset 2023 from the Canadian Institute for Cybersecurity (*Neto et al., 2023*). The dataset contains the traffic flow of 34 types of IoT classes. Experiments are performed using the feature selection technique and the proposed model. Experimental results are given in Table 6. The results indicate that the proposed model can perform well on other datasets as well.

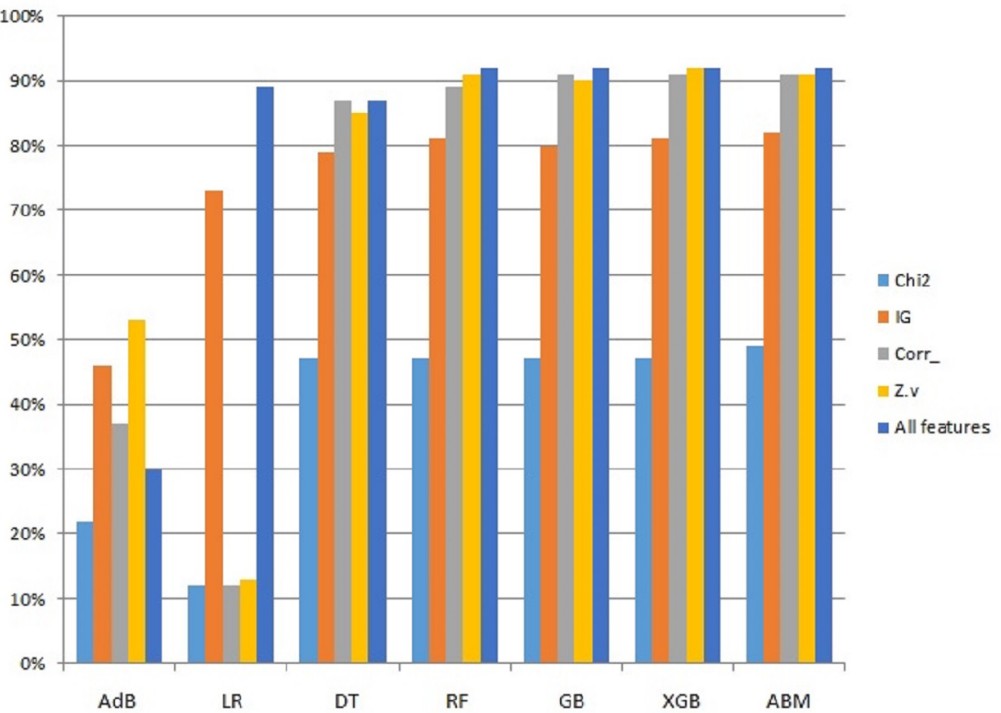

**Figure 5** Graphical representation of accuracy results using ABM and features selection.

**Table 6** Experimental results for the IoT Dataset 2023.

| Algorithms | Precision | Recall | Accuracy | CV |
|---|---|---|---|---|
| ABM | 97.8 | 98.3 | 99.8 | 99 |

**Table 7** Performance comparison with existing studies.

| Reference | Approach | Accuracy |
|---|---|---|
| *Hamad et al. (2019)* | Machine Learning | 90.3% |
| *Aksoy & Gunes (2019a)* | SysID | 82% |
| *Kawai et al. (2017)* | SVM | 88.3% |
| Proposed | Ensemble | 91% |

## Performance comparison with other studies

The results of the proposed approach are compared with several existing approaches that perform IoT device identification. For example, *Hamad et al. (2019)* used 67 features for IoT device identification and reported an accuracy of 90.3%. Similarly, *Aksoy & Gunes (2019a)* proposes an approach SysID to identify IoT devices using a single packet and obtains an accuracy of 82%. In comparison, the proposed approach shows better performance for IoT device identification, as shown in Table 7.

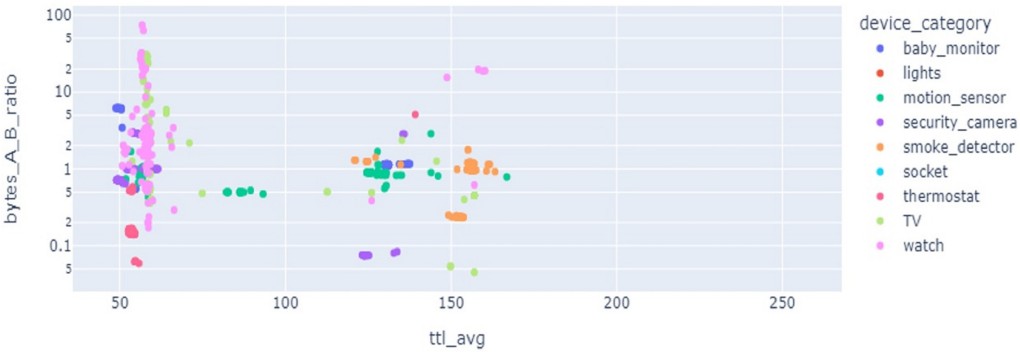

**Figure 6** Feature space for different kinds of IoT devices.

## Discussions

Results of LR show precision 92%, recall 91%, and accuracy 91% but when cross-validation is applied its accuracy score goes down to only 11% as the given data classes are overlapped and have highly correlated features. SVC results for precision, recall, and valid accuracy are 86%, 85%, 85%, and 18%. SVC shows good results in cases where data points have maximum marginal distance and hyperplane can distinguish between different classes. Being highly correlated data points overlapped, the marginal distance can not be identified and a clear hyperplane can not be drawn, so, its accuracy goes down. Also, the data is not present in a linear form which also degrades the results. As SVC is sensitive to the outlier, if the model is trained with training data having maximum outlier, its performance is worse. Feature selection is important, as multiple devices can have overlapping features which affects the classification performance. Figure 6 reveals that the features from multiple devices overlap, thereby making the classification difficult.

AdB is good for quality data and does not show good results for noisy data and data containing outliers. AdB works on the principle of series prediction and for the current data, it does not produce good results with a validation accuracy of 51%. KNN makes the group of data points concerning some near-defined center point and is also affected by outliers. For the current dataset, it shows moderate results with 87% precision, 87% recall, 87% accuracy, and a cross-validation score of 80%. DT shows good results with a precision of 89%, recall of 88%, accuracy of 88%, and validation accuracy of 87%. XGB, RF, and GBM classifiers show significant results in terms of precision, recall, accuracy, and cross-validation scores of up to 91%, 91%, 91%, and 90%, respectively.

The proposed ABM models show better results than all the employed models and obtain a precision of 92%, recall of 91%, F1-score of 91%, accuracy of 91%, and a validation score minimum of 89% and maximum of 95%. The average validation score is 91% which is higher than other models. The proposed ABM model uses the hard voting criterion to give the final prediction.

This study used 'RepeatedStartifiedKFold' cross-validation method to measure the expediency and over-fitting of the proposed model. An overfitted model shows very poor results and such a model does not have much value to predict new events. We checked

**Table 8    Performance comparison of machine learning models.**

| Algorithms | Precision | Recall | Accuracy | CV |
|---|---|---|---|---|
| LR | 92 | 91 | 91 | 11 |
| SVC | 86 | 85 | 85 | 18 |
| AdaB | 58 | 61 | 61 | 49 |
| KNN | 87 | 87 | 87 | 80 |
| DT | 88 | 88 | 87 | 87 |
| GDB | 91 | 91 | 91 | 90 |
| RF | 92 | 92 | 92 | 90 |
| Xgboost | 91 | 91 | 91 | 90 |
| ABM | 92 | 91 | 91 | 91 |

the authenticity of the proposed model with 2, 3, 5, and 10 n_splits and n_repeats =3 cross-validation tests. The reactions with 10-split cross-validation scores are much more significant as shown in Table 8.

Feature selection techniques like Chi2, correlation, zero variance, information gain, and all features are also adopted to check the importance of given features, and their performance is discussed. Dropping constant features results based on 253 features RF, GDB, and XGB show an accuracy of 90%. Using the top 10 features from Chi2 features selected based on lesser *p*-values among all features showed poor results. Another feature selection method mutual information gain was also used to obtain the top 10 features where RF, GDB, and XGB show an accuracy of 82% while the proposed model ABM shows 83%. Similarly, correlation-based feature selection was also considered in which RF, GDB, and XGB show an accuracy of 90% and ABM shows 91% accuracy. The confusion matrix for the proposed approach is shown in Fig. 7 indicating the superior performance for each type of IoT device except for socket which has a true positive score of 0.7.

Despite the good performance of the proposed approach, for some IoT devices, the performance is under par. For example, the socket class has a 0.3 false prediction score and needs to be improved to further increase the accuracy of device identification. Similarly, the lights class has a 0.8 true prediction score which can be further improved. Current experiments involve only nine devices, and further devices can be added to investigate the performance of the proposed approach.

## CONCLUSION

Network security has a great challenge to monitor connected device traffic and newly installed IoT devices in the existing networks as the attacker can approach connected devices using the vulnerabilities. In this study, we present a classification approach to identify known and unknown IoT devices connected to smart homes or offices with the local networks. Feature engineering techniques have also a great contribution to inspecting the traffic of network-connected devices and have a significant impact on the results to identify the IoT devices. Several types of feature engineering approaches are evaluated in this study for IoT device identification. The proposed ABM model provides better results with 91% accuracy by combining the results of several individual algorithms based on the

**Figure 7** Confusion matrix for the proposed approach.

features extracted from the web traffic of IoT devices. The proposed model is beneficial for the network administrators to manage the connected device and also ensure security management for intruder devices.

### Funding

This work was supported by the Basic Science Research Program through the National Research Foundation of Korea (NRF) funded by the Ministry of Education (NRF-2019R1A2C1006159). There was no additional external funding received for this study. The funders had no role in study design, data collection and analysis, decision to publish, or preparation of the manuscript.

### Grant Disclosures

The following grant information was disclosed by the authors:
Basic Science Research Program through the National Research Foundation of Korea (NRF) funded by the Ministry of Education: NRF-2019R1A2C1006159.

## Competing Interests

Imran Ashraf is an Academic Editor for PeerJ Computer Science.

## Author Contributions

- Sajjad Hussain conceived and designed the experiments, performed the experiments, performed the computation work, prepared figures and/or tables, and approved the final draft.
- Waqar Aslam performed the experiments, performed the computation work, prepared figures and/or tables, and approved the final draft.
- Arif Mehmood performed the experiments, performed the computation work, prepared figures and/or tables, and approved the final draft.
- Gyu Sang Choi conceived and designed the experiments, analyzed the data, authored or reviewed drafts of the article, and approved the final draft.
- Imran Ashraf conceived and designed the experiments, analyzed the data, authored or reviewed drafts of the article, and approved the final draft.

## Data Availability

The data is available at Kaggle:

Available at https://www.kaggle.com/code/fanbyprinciple/iot-device-identification-using-machine-learning/input

The code is available at Zenodo: Sajjad Hussain. (2023). IoT Device Identification Using Web Traffic. Available at https://doi.org/10.5281/zenodo.8416098.

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
