# Peer review of "A machine learning based framework for IoT devices identification using web traffic"

_PeerJ Computer Science, doi:10.7717/peerj-cs.1834_

## Round 0.1 · original submission · Major Revisions

According to the reviewers, the manuscript needs a revision before acceptance.

**Language Note:** The review process has identified that the English language must be improved. PeerJ can provide language editing services - please contact us at copyediting@peerj.com for pricing (be sure to provide your manuscript number and title). Alternatively, you should make your own arrangements to improve the language quality and provide details in your response letter. – PeerJ Staff

Reviewer 1 ·

Basic reporting

The abstract mentions the use of machine learning models, feature engineering, and cross-validation but lacks specific details about the methodologies and techniques employed. Providing more information on the approach used would help readers understand the study's rigor and potential limitations.
The abstract does not mention the dataset used for training and testing the ABM. It's essential to know the source, size, and diversity of the dataset to assess the model's generalizability and potential bias.
The paper highlights impressive performance metrics (91% accuracy, 93% precision, recall, and F1 score), it's crucial to address the generalizability of these results. Are these results specific to a particular setup, or can they be applied to a broader range of IoT devices and network configurations?
The introduction is quite dense and contains a significant amount of information without clear separation of ideas. The reader might struggle to follow the flow of the text, and some sentences are convoluted, making it challenging to grasp the main points.
While citing previous research is essential for providing a foundation for the study, the introduction contains numerous citations that interrupt the flow of the text. A more concise and focused citation approach would improve readability.
Some points are repeated throughout the introduction, particularly the emphasis on data privacy concerns and the importance of device identification. This redundancy could be condensed to make the text more concise.
Although the paper highlights the importance of IoT device identification, it lacks a precise problem statement that defines the research gap or specific challenges the article aims to address. A clear problem statement would help readers understand the focus of the study.

Experimental design

Needs a lot of improvement, such as complete dataset description, data analysis, classes of dataset etc

Validity of the findings

The finding are valid but not enough, author need to add more comparsion (atleast 1) with recent works based on some crucial factor.

Additional comments

NIL

·

Basic reporting

The paper titled "A machine learning based framework for IoT devices identification using web traffic", is a novel writing. This paper purposed an accuracy boosting model (ABM) using machine learning algorithms of random forest and extreme gradient boosting. Featuring engineering techniques are employed along with cross-validation to accurately identify IoT devices such as lights, smoke, detector, thermostat, motion-sensor, baby-moitr, socket, TV, security-camera, and watch. The paper is clearly written in a good style and includes figures and tables wherever necessary.

Experimental design

Experimental design:
The authors have clearly acknowledged and identified the contributions of their research against previous researchers' work
The purpose of the paper has been very well stated in the abstract but needs clarification on the following:
1. What is the motivation for choosing ensemble Models for this particular task?
2. In the discussion section, the research's strengths, limitations, and generality need more appropriate discussions.

Validity of the findings

The authors adequately evaluated their work, and all claims are clearly articulated and supported by empirical experiments.

Additional comments

No

---

## Round 0.2 · accepted · Accept

The authors correctly addressed the issues raised in the first review round.

Reviewer 1 ·

Basic reporting

The authors prepared the manuscript according to the previously listed comments. I am happy to suggest the acceptance of the article.

Experimental design

The authors prepared the manuscript according to the previously listed comments. I am happy to suggest the acceptance of the article.

Validity of the findings

The authors prepared the manuscript according to the previously listed comments. I am happy to suggest the acceptance of the article.